

# Effect of kinematics on ground reaction force during single-leg jump landing in children: a causal decomposition approach in jumpers and non-jumpers

Carlos Cruz-Montecinos[1], Isaac Estevan[2], Jiri Skypala[3], Claudio Tapia-Malebrán[4,5] and Xavier García-Massó[6]

[1] Department of Physical Therapy/Faculty of Medicine, University of Chile, Santiago, Chile
[2] AFIPS Research Group, Department of Physical Education, Arts and Music, University of Valencia, Valencia, Spain
[3] Department of Human Movement Studies/Human Motion Diagnostic Centre, University of Ostrava, Ostrava, Czech Republic
[4] Department of Physical Therapy, Faculty of Medicine, University of Chile, Santiago, Chile
[5] Department of Physical Therapy, Catholic University of Maule, Talca, Chile
[6] HUMAG Research Group, Department of Physical Education, Arts and Music, University of Valencia, Valencia, Spain, Universidad de Valencia, Valencia, Spain

Corresponding author
Carlos Cruz-Montecinos,
carloscruz@uchile.cl

## ABSTRACT

**Background:** The interaction between joint kinematics and kinetics is usually assessed by linear correlation analysis, which does not imply causality. Understanding the causal links between these variables may help develop landing interventions to improve technique and create joint-specific strengthening programs to reduce reaction forces and injury risk.

**Objective:** Therefore, the aim of this study was to analyze the causal interaction between lower limb sagittal kinematics and vertical ground reaction force (VGRF) during single-leg jump landing in children who are jumpers (volleyball and gymnastics) and non-jumpers, using the causal empirical decomposition method. Our hypothesis is that children who participate in jumping sports, compared to those who do not, employ a different joint strategy to regulate ground reaction forces during landing, particularly at the ankle level.

**Methods:** Two groups were compared: the jumpers group ($n = 14$) and the non-jumpers (control group, $n = 11$). The causal interaction between sagittal kinematics and VGRF was assessed using ensemble empirical mode decomposition (EEMD) and time series instantaneous phase dependence in bi-directional causality. The relative causal strength (RCS) between the time series was quantified as the relative ratio of absolute cause strength between kinematics and VGRF.

**Results:** A significant interaction between joint and group was found for RCS ($p = 0.035$, $\eta^2 p = 0.14$). The *post-hoc* analysis showed the jumpers group had higher ankle-to-VGRF RCS than the control group ($p = 0.017$, $d = 1.03$), while in the control group the hip-to-VGRF RCS was higher than the ankle-to-VGRF RCS ($p = 0.004$, $d = 0.91$).

**Conclusion:** Based on the causal decomposition approach, our results indicate that practicing jumping sports increases the causal effect of ankle kinematics on ground reaction forces in children. While non-jumper children rely more on the hip to modulate reaction forces, jumper children differ from non-jumpers by their greater

use of the ankle joint. These findings could be used to develop specific training programs to improve landing techniques according to practice level, potentially helping to reduce the risk of injury in both athletes and non-athletes.

## INTRODUCTION

Improving motor competence in childhood and adolescence is key to individuals' integral development (*i.e.*, physical activity participation and fitness) (*Stodden et al., 2008*) and fosters the adoption of healthy lifestyles in the short and long term (*Barnett et al., 2022*). Additionally, motor competence can be a prevention factor in sports-related injuries, since certain coordination patterns have been described as risk factors in several non-contact musculoskeletal injuries (*Pfeifer et al., 2019*). Assessing the impact forces during jumping and landing is essential for understanding how expertise in motor actions modulates biomechanics and injury risk (*Devita & Skelly, 1992*; *Caulfield & Garrett, 2004*; *Aerts et al., 2013*). Several studies have found a linear relation between kinematic parameters and kinetic variables (*i.e.*, vertical ground reaction force (VGRF)) to identify the coordination patterns that minimize impact forces (*Devita & Skelly, 1992*; *Yu, Lin & Garrett, 2006*; *Hoch et al., 2015*). Based on linear approaches analysis, the role of the knee joint and ankle in absorbing the impact on landing has been highlighted (*Norcross et al., 2010*; *Fong et al., 2011*; *Martinez et al., 2022*). However, it has been reported that the lower extremity coordination presents a non-linear trajectory (*i.e.*, exponential and natural logarithmic relationships) during the landing phase (*Yeow, Lee & Goh, 2011*). Additionally, the correlation between joint kinematics and kinetics does not necessarily imply causality between the variables (*Wille et al., 2014*; *Tamura, Akasaka & Otsudo, 2021*; *Martinez et al., 2022*).

In young athletes, overuse injuries in jumping sports are common due to factors such as previous injuries, family history of musculoskeletal disorders, absence from regular training, and high training loads (*Leppänen et al., 2017*). Competitive gymnastics often involve high-impact landings compared to recreational gymnastics, raising concerns about repetitive stress injuries because of high-impact loading (*Seegmiller & McCaw, 2003*). A stiffer jump-landing technique has been reported as a risk factor for acute injuries and the development of overuse injuries, as it involves reduced active motion in the lower extremities (*Aerts et al., 2013*). *Estevan et al. (2020)* found that children who practiced volleyball showed lower impact forces and higher knee flexion during single-leg landings than those who practiced gymnastics. Despite these findings, the causal interaction between joint kinematics and VGRF, particularly during landing from jumping tasks, remains unexplored. Unlike traditional linear correlation analyses, nonlinear methods better capture the complexity of interactions between joint movements and reaction forces, which often do not follow simple linear patterns. One of the main limitations of the linear approach is its inability to provide specific information about the direction of causality and

the strength of causal interactions between joint motion and reaction forces. Both are crucial for understanding how different landing techniques modulate reaction forces. Analyzing these causal interactions in a joint-specific manner can lead to targeted joint-strengthening programs that optimize landing techniques and prevent injuries.

To address these limitations, non-linear methods may offer a more effective approach to representing the physical nature (*i.e.*, nonlinear and non-stationary) of the data and the causal interactions between kinematic and kinetic signals. For instance, *Yang, Peng & Huang (2018)* proposed the causal empirical method, which determines causality by identifying changes in instantaneous phase dependency between signals when a component causally related to another is removed. This method is based on the principle that a cause (input) produces an effect when present, and the effect diminishes or disappears when the cause is removed. Thus, A causes B if the intrinsic component in B that is causally related to A is removed from B, and as a result, the instantaneous phase dependency between A and B diminishes. This leads to the conclusion that A causes B, and not the other way around (*Yang, Peng & Huang, 2018*). This method is based on cause-and-effect covariation, measured through instantaneous phase coherence, rather than prediction or temporal dependence (*Yang, Peng & Huang, 2018*).

The causal decomposition approach can be particularly interesting for the assessment of causal interactions between kinematics and kinetics during landing motor tasks. However, even though causal empirical decomposition has been proven by ecological data, neurophysiology and biomechanics during gait (*Zhang et al., 2019*; *Peng et al., 2021*; *Cruz-Montecinos et al., 2022*), it has yet to be proven in biomechanical studies during jump landing. Therefore, the aim of this study was to analyze the causal interaction between lower limb sagittal kinematics and VGRF during single-leg jump landing in children who are jumpers (volleyball and gymnastics) and non-jumpers, using the causal empirical decomposition method. Our hypothesis is that children who participate in jumping sports, compared to those who do not, employ a different joint strategy to regulate ground reaction forces during landing, particularly at the ankle level. Understanding the causal relationships between these variables could help improve landing training programs for children with and without jump training experience.

## MATERIALS AND METHODS

### Participants

Twenty-five children complied with the inclusion criteria, as follows: (i) regular practice of volleyball ($n = 8$) or gymnastics ($n = 6$) for at least one year (jumpers group) or no practice of any structured sport in their leisure time ($n = 11$, non-jumpers); (ii) no injuries in the 6 months prior to the tests; and (iii) have body mass index below 25 kg/m$^2$. For the jumpers group, the minimum duration of sports practice was set at 1 year and they trained at least twice a week. No differences in age, body mass index or % of body fat between the jumpers and non-jumpers (control group) were found (Table 1). The study was approved by the University Valencia Ethics Committee (H1446557620395) and the parents of the participants gave their written informed consent. Some of the data given here have been

**Table 1 Comparison of characteristics between groups.**

|  | Non-Jumpers (*n* = 11) | Jumpers (*n* = 14) | *p*-value | Effect size |
|---|---|---|---|---|
| Age (years) | 9.1 [8.9 to 12.9] | 9.8 [7.9 to 12.9] | 0.475 | *r 0.17* |
| Weight (kg) | 37.9 [22.0 to 55.0] | 28.5 [22.6 to 65.7] | 0.171 | *r 0.33* |
| Height (cm) | 142 ± 11.1 | 136.2 ± 12.3 | 0.199 | *d 0.53* |
| BMI (kg/m$^2$) | 19.3 [14.1 to 24.1] | 16.2 [14.6 to 23.1] | 0.139 | *r 0.36* |
| Body fat (%) | 18.7 ± 7.6 | 14.3 ± 12.7 | 0.147 | *d 0.60* |
| Girls/boys | 6/5 | 4/10 | 0.188 | *v 0.26* |
| Practice (years) | —— | 3 [1 to 7] | N.A |  |

**Note:**

Data are presented as mean ± standard deviation (SD) for normally distributed variables and as median (range) for non-normally distributed variables. The *p*-value indicates the significance of the difference between groups. Effect sizes (in italics) are reported as *r* (Rank Biserial Correlation) and *d* (Cohen's d) for continuous variables, and *v* (Cramér's V) for categorical variables. BMI, body mass index; N.A, not applicable.

used in a previous article (*Estevan et al., 2020*), which addressed a different research question using another data analysis method.

## Apparatus

VGRF and joint angles data of the lower limb during landing were acquired using a nine-camera motion capture system (Oqus, 240 Hz; Qualisys, Gothenburg, Sweden) synchronized with a force plate (9281CA, 1,200 Hz; Kistler, Winterthur, Switzerland). Reflective markers were securely positioned to define segments (*Estevan et al., 2020*). Data from the markers were processed by Visual 3D software (C-Motion, Germantown, MD, USA).

## Procedure

After a standardized individual warmup (including familiarization), the barefoot participants performed 10 single-leg landings from a height of 25 cm onto the force plate, keeping their hands on their hips during the task (Fig. 1A). The take-off platform was located 5 cm behind the force platform. The dominant leg was the one used for landing (kicking preference). The child, acting on their own decision, took a step with the ground leg, landing on the force platform. All children were encouraged to land on one leg while maintaining stability for 10 s. The free leg was kept in the rear with the knee flexed 90° approximately. The trials were video recorded to reject those who started jumping instead of stepping or the free knee was not flexed around 90°. The trials started preferably between 1–3 s after the research assistant gave the appropriate signal and a rest interval of 1–2 min was allowed between the trials.

## Signal processing and causal interaction assessment

The data from 10 single-leg landing tests were used for further analysis. VGRF data were smoothed with a fourth-order 75 Hz low-pass Butterworth filter (*Blackburn et al., 2016*), and the same filter was applied to the sagittal hip, knee and ankle joint angle to preserve the dynamic consistency between signals (*Tomescu et al., 2018*) (Fig. 1B). Jump landing signals (*i.e.*, landing cycle) were determined based on a threshold VGRF value of 15 N to the

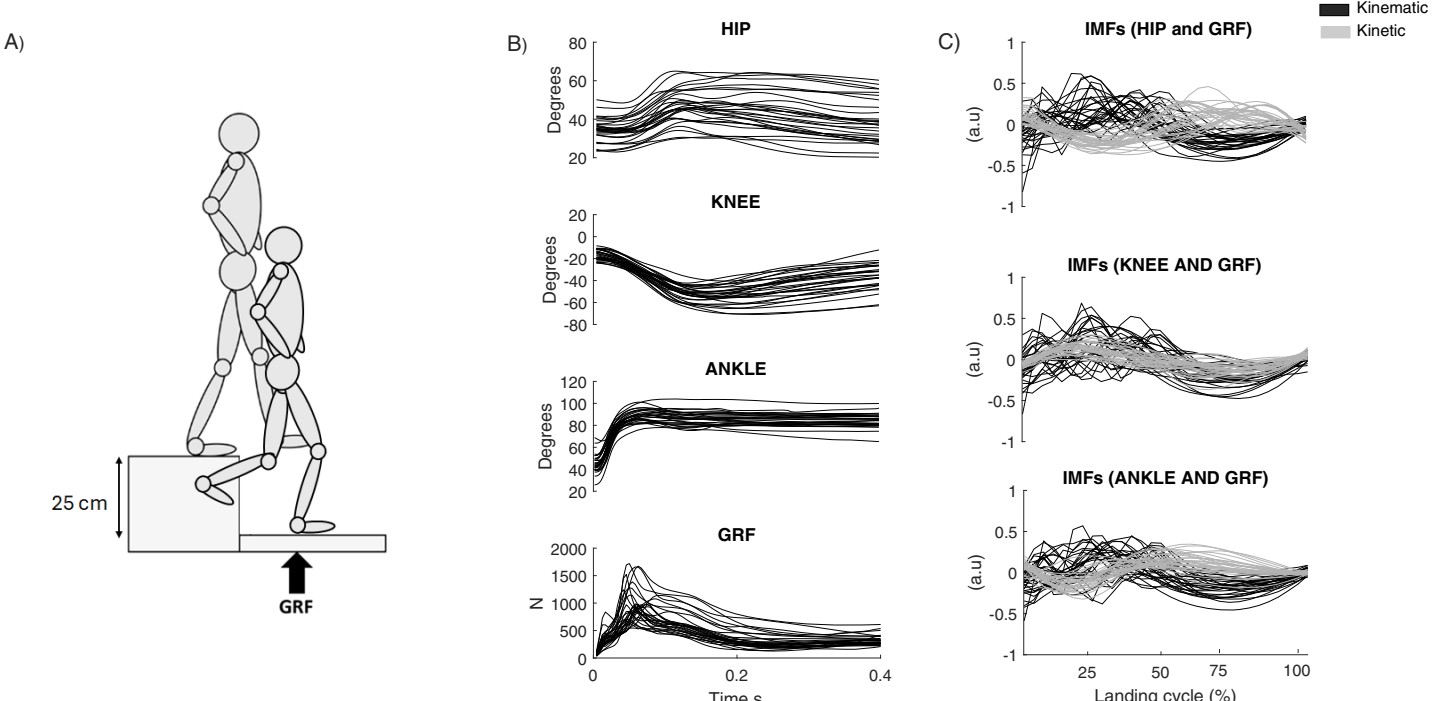

**Figure 1** (A) Schematic representation of a single-leg jump landing. (B) Average kinematics and vertical ground reaction force (VGRF) for 10 cycles for each participant (*n* = 25). (C) Average of the intrinsic mode function (IMF) selected for each participant (*n* = 25).

maximum knee flexion on the dominant lower limb. The causal interaction between the sagittal joint kinematics and VGRF on the landing cycle was assessed by ensemble empirical mode decomposition (EEMD) and time series instantaneous phase dependence in bi-directional causality, using the Yang methodology (*Yang, Peng & Huang, 2018*). The code for its calculation can be downloaded from the study by *Yang, Peng & Huang (2018)*. EEMD with adapted additive noise was used to obtain the intrinsic mode functions (IMFs). Unlike wavelet decomposition or Fourier transform, EEMD is a self-adaptive and data-driven technique that does not need a pre-set basis function (*Chan et al., 2014*; *Peng et al., 2021*). Based on previous methodology, white noise was added to enhance the separability of IMFs during decomposition (*Chan et al., 2014*; *Cruz-Montecinos et al., 2022*). Briefly, in each landing test, the EEMD was used to obtain a finite number of IMFs and identify the instantaneous phase coherence between paired IMFs. The noise level with the lowest root-mean-square was selected for further analysis by EEMD. This step is critical to avoiding spurious causal detection resulting from the poor separation of a given signal. Based on Yang's methodology, the relative causal strength (RCS) was then quantified as the relative ratio of absolute strength between IMFs (*Yang, Peng & Huang, 2018*). The maximal RCS was utilized to evaluate both the intensity and the direction of the causal relationships between the sagittal joint angles and the VGRF. An RCS ratio of 0.5 suggests an absence of causality, whereas ratios close to 0 or 1 indicate a strong causal influence of the corresponding IMFs in the bidirectional interactions (*Yang, Peng & Huang, 2018*). In our case, it indicates a strong causal influence between the kinematic data

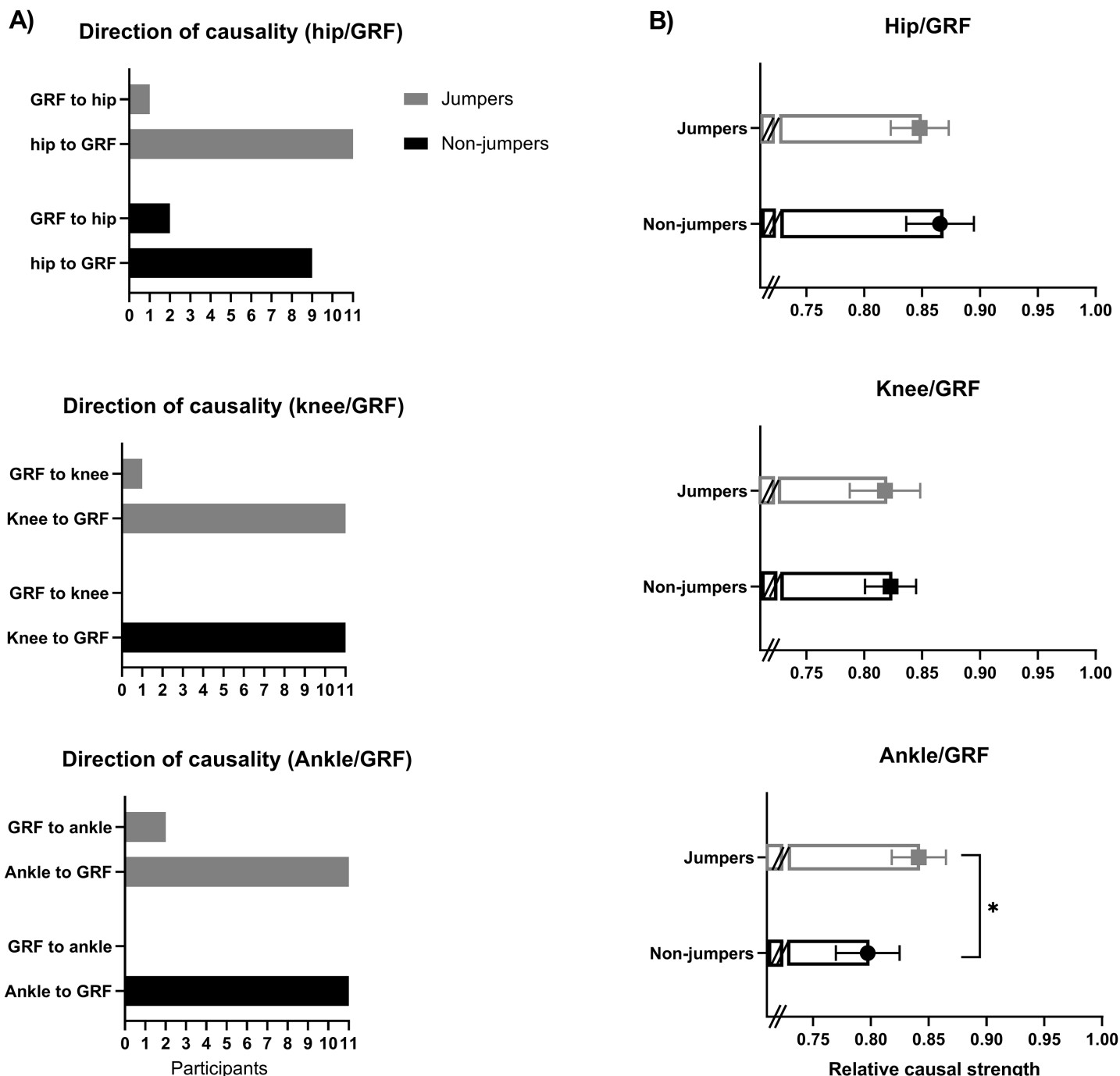

**Figure 2 Direction of causality and relative causal strength.** (A) Direction of causality between groups (kinematic → VGRF or VGRF → kinematic). (B) Ground reaction force (GRF). The relative causal strength (RCS) between groups. RCS are expressed as a mean with a 95% confidence level. *<0.05.

and VGRF (*e.g.*, from kinematics to VGRF or vice versa). For example, an RCS close to 1 between the sagittal joint angles and VGRF suggests that changes in joint kinematics are significantly impacting the forces transmitted through the foot during landing (kinematic → VGRF). Figures 1B and 1C show the kinematic and kinetic signals, and the selected IMFs

with higher RCS based on the landing cycle, respectively. The median IMF frequency was used to characterize the frequency of IMFs selected for the sagittal joint kinematics and VGRF.

## Statistical analysis

All the data were analyzed on SPSS (IBM, Armonk, NY, USA). A $p$-value of $<0.05$ was considered statistically significant. Normality was assessed using the Shapiro-Wilk test. Age, body mass, and body mass index showed non-normal distributions. The RCS of the hip in the non-jumpers group and the median frequency of the IMF for ankle kinematics also exhibited non-normal distributions. The chi-squared test was used to compare the distribution of the direction of causality between joints and groups. To compare the RCS and the median frequency of the IMFs selected between joints and groups, a mixed ANOVA (joints × groups) were applied, followed by pairwise comparisons with Bonferroni correction. A $p$-value of $<0.05$ was considered statistically significant. Even though the hip RCS and median frequency of the IMF for ankle kinematics in the non-jumpers group were not normally distributed, the mixed ANOVA assumptions were met. Mauchly's test confirmed sphericity ($p = 0.792$ and $p = 0.977$), and Levene's test showed equal variances for both RCS and median frequency ($p > 0.05$). Finally, the effect sizes were calculated for all variables. For ANOVA, partial eta squared ($\eta^2 p$) was used to interpret effect sizes, categorized as follows: trivial ($\eta^2 p < 0.01$), small ($0.01 \leq \eta^2 p < 0.06$), moderate ($0.06 \leq \eta^2 p < 0.14$), and large ($\eta^2 p \geq 0.14$). For normally distributed variables, Cohen's d was used: trivial ($d < 0.2$), small ($0.2 \leq d < 0.5$), moderate ($0.5 \leq d < 0.8$), and large ($d \geq 0.8$). For non-normally distributed variables, the Rank Biserial Correlation ($r$) was used: trivial ($r < 0.1$), small ($0.1 \leq r < 0.3$), moderate ($0.3 \leq r < 0.5$), and large ($r \geq 0.5$). For categorical variables, Cramér's V was used: trivial ($v < 0.1$), small ($0.1 \leq v < 0.3$), moderate ($0.3 \leq v < 0.5$), and large ($v \geq 0.5$).

# RESULTS

The average of sagittal joint angles kinematics VGRF and IMF selected for 10 cycles for each participant are shown in Figs. 1B and 1C, respectively.

## Direction of causality

There were no differences between groups in the direction of causality (hip $p = 0.399$, $v = 0.17$; knee $p = 0.366$, $v = 0.18$; ankle, $p = 0.191$, $v = 0.26$) (Fig. 2A). The direction of causality from kinematics to VGRF (*i.e.*, kinematics → VGRF) was observed in 92% of participants in the ankle joint, 88% in the hip joint and 96% for knee joint (Fig. 2A). The consistency in the direction of causality (average of 92%), along with the high RCS value (average of 0.83), confirms the influence of kinematics on VGRF.

## Strength of causality

For RCS, the mixed ANOVA showed a main significant effect of joint factor ($F_{(2,46)} = 6.30$, $p = 0.004$, $\eta^2 p = 0.22$), with no differences between the groups ($F_{(1,23)} = 0.42$, $p = 0.524$, $\eta^2 p = 0.02$). A significant interaction between joint × group ($F_{(2,46)} = 3.60$, $p = 0.035$, $\eta^2 p = 0.14$) was found. The *post-hoc* analysis showed that the RCS from hip-to-VGRF was

higher than ankle-to-VGRF ($p = 0.004$, $d = 0.91$) in the control group, while no differences were observed between RCS joints in the jumpers group ($p > 0.05$). Notably, the jumpers group had higher ankle-to-VGRF RCS compared to non-jumpers (($p = 0.017$, $d = 1.03$); Fig. 2B). For all *post-hoc* comparisons and effect size see Table S1. Overall, these results indicate that non-jumping children use the hip more to modulate reaction forces during landing, while jumping children rely more on the ankle joint compared to non-jumpers to regulate these forces.

### Characteristics of IMF selected

The median IMF frequency selected from joint kinematics was found between 16–17 Hz and between 17–21 Hz from VGRF. The mixed ANOVA for IMFs from kinematics and VGRF showed a non-significant main effect for joint ($F_{(2,46)} = 0.09$, $p = 0.916$, $\eta^2 p = 0.01$ and $F_{(2,46)} = 1.42$, $p = 0.251$, $\eta^2 p = 0.06$, respectively), non-significant main effect for group ($F_{(1,23)} = 0.04$, $p = 0.842$, $\eta^2 p = 0.01$ and $F_{(1,23)} = 0.31$, $p = 0.586$, $\eta^2 p = 0.01$, respectively), and non-significant joint x group interaction effect ($F_{(2,46)} = 1.07$, $p = 0.352$, $\eta^2 p = 0.04$ and $F_{(2,46)} = 0.57$, $p = 0.569$, $\eta^2 p = 0.02$, respectively).

## DISCUSSION

Based on causal decomposition approach, the main results of our study indicated that (i) the direction of causality during landing was primarily from joint kinematics to VGRF, confirming the causal approach used in our study; (ii) in the non-jumpers group, there was a predominance of hip kinematics to VGRF interaction, whereas the ankle-to-VGRF causal relationship was higher in the jumpers group compared to the non-jumpers group. These results support our hypothesis that children who participate in jumping sports, compared to those who do not, employ a different joint strategy to regulate ground reaction forces during landing, particularly at the ankle level. Our results align with previous studies that have examined the role of ankle joint kinematics on ground reaction forces during landing (*Taylor et al., 2022*; *Tait et al., 2022*). To the best of our knowledge, this is the first study to confirm, using the causal decomposition approach, that jumper and non-jumper children modulate ground reaction forces through different joint strategies. The differences observed could guide joint-specific strengthening programs aimed at attenuating VGRF and potentially reducing injury risk.

Notably, our study not only differentiates the causal strength between kinematic and kinetic factors between the groups but also highlights the distinct strategies adopted during landing. The higher RCS (close to 1) between ankle kinematics and VGRF observed in jumpers compared to non-jumpers (differences with large effect size, $d = 1.03$) suggests that jumpers have superior control at the ankle level to modulate the forces transmitted during landing. In contrast, non-jumpers exhibited a higher RCS at the hip compared to the ankle (with a large effect size, $d = 0.91$), indicating a different landing strategy than that of the jumpers. Jumping landing exercises in the daily routine of jumping sports may be a key factor in the differences observed between the groups in this study. Landing strategies used by the jumpers group could thus be changed by using ankle kinematics to promote a safer motor strategy to reduce bone bending moments, muscle forces and VGRF, as has

previously been suggested (*Mills, Pain & Yeadon, 2009*). The ankle is the first major joint to absorb energy during landing (*Fong et al., 2011*; *Martinez et al., 2022*). The ankle strategy in children who practice jumping sports may be a motor strategy adopted to reduce the VGRF. Different studies have noted the clinical relevance of joint motion on the effect of reaction force during landing, particularly at the ankle joint level (*Devita & Skelly, 1992*; *Fong et al., 2011*; *Hoch et al., 2015*; *Martinez et al., 2022*; *Tayfur et al., 2022*; *Tait et al., 2022*). The differences observed may result from the neuromuscular strategy employed by the muscles during landing (*Kim, Palmieri-Smith & Kipp, 2020*). Improved motor actions of the quadriceps, gluteus, and soleus muscles enhance eccentric work, which is crucial for joint limb control and landing stability (*Maniar et al., 2022*). These muscles are commonly involved during jump sports practice and may be key in enhancing the interaction between ankle motion and VGRF. Regarding the characteristics of the selected IMFs, we found that the frequencies of the kinematics and VGRF ranged between 15–19 Hz, showing similar frequencies between groups. Future studies are needed to better understand the causal interaction between muscle coordination and jump landing, particularly the causal interaction between kinematics and VGRF.

## Practical implications

These findings could lead to training programs which adjust landing strategies to minimize lower limb injuries and improve motor function in young athletes. Understanding the causes and effects of joint kinematics and VGRF allows the creation of specific training programs to optimize landing strategies and decrease the risk of injuries. Promoting a greater range of ankle motion could help lower VGRF, thereby reducing injury risk by enhancing shock absorption (*Taylor et al., 2022*; *Tait et al., 2022*; *Xu et al., 2024*). By improving both ankle flexibility and strength, athletes may decrease the risk of overload and soft tissue injuries (*Hopper et al., 2017*; *Martinez et al., 2022*; *Maniar et al., 2022*). According to our results, ankle movement seems to be more relevant for jumpers, while hip movement is more significant for non-jumpers in managing landing forces. Therefore, prioritizing ankle strengthening for jumpers and hip strengthening for non-jumpers appears to be an approach that could potentially optimize the modulation of reaction forces during landing. As indicated in Fig. 3, progression in training could involve transitioning to integrated ankle and hip strengthening as the landing technique improves. Furthermore, the increased causal interaction from hip kinematics to VGRF, compared to ankle kinematics to VGRF, may indicate an improvement in landing technique based on athletes' progress. However, future studies are needed to corroborate these assumptions in longitudinal research. Additionally, future studies could be conducted to understand the effect of neuromuscular training (*e.g.*, plyometric and joint-specific strengthening programs) on the causal interaction between kinematics and kinetics in non-jumper and jumper young athletes (*Hopper et al., 2017*). Furthermore, future studies are needed to understand the effect of training programs and different types of feedback to determine whether these interventions can modify the causal interaction between kinematic and kinetic forces (*Prapavessis & McNair, 1999*; *Aerts et al., 2013*), particularly at the ankle joint, in order to restore more normal patterns of force absorption and reduce the

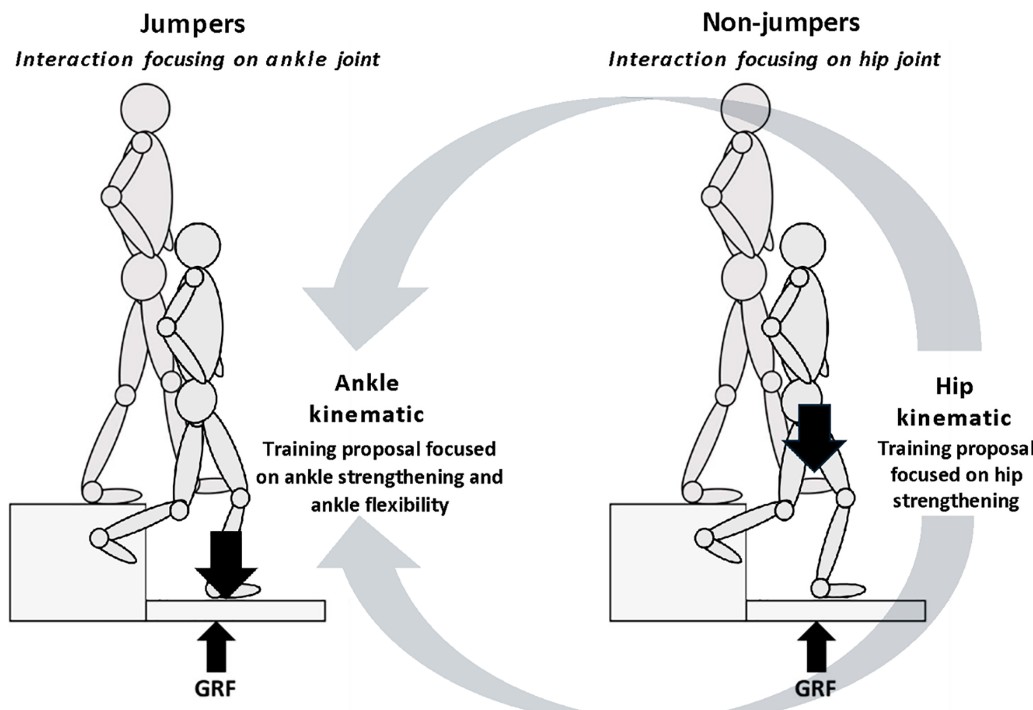

**Figure 3 Kinematic-GRF interaction differences and proposed training progression across groups.** The difference in interaction between kinematic and ground reaction force (GRF) was predominantly at the ankle between groups. In the control group, this interaction primarily occurred at the hip joint. This figure illustrates the potential progression in training focus from ankle to hip, or integrating both, depending on the athlete's experience and skill development.

occurrence of repeated micro-trauma to ankle structures, especially in children with a history of ankle injuries (*e.g.*, ankle sprain or restricted dorsiflexion) (*Caulfield & Garrett, 2004*; *Taylor et al., 2022*).

Unlike linear approaches, nonlinear methods, such as causal empirical decomposition, provide a more specific approach based on the physical nature (*i.e.*, nonlinear and non-stationary data) of the kinetic and kinematic signals during landing. It has been reported that the lower extremity coordination presents a non-linear interaction during the landing phase (*Yeow, Lee & Goh, 2011*). Our study suggests that non-linear analysis based on causal decomposition is a valuable method for assessing the direction and strength of causal interactions between kinematics and kinetics in biomechanical landing research. It may also contribute to the study of other motor tasks.

Our study has several limitations that should be considered: (i) the landing strategy was assessed from a single height. The VGRF peaks and eccentric muscle work by the lower extremity joints may increase with increased landing heights (*Zhang et al., 2008*). (ii) It was not possible to associate the results obtained with training load experience or to evaluate the effect of jumping practice as a pre- and post-methodology. Future longitudinal studies

are needed to investigate the effect of jump training on causal relationships between kinematics and VGRF. (iii) The muscle activity during the landing task was not collected; therefore, it is not possible to determine if there was a different neuromuscular strategy associated with the differences observed. (iv) The maturity offset was not assessed in our study. With maturation, male adolescent athletes report increased vertical jump height and lower landing forces, whereas female athletes have not (*Quatman et al., 2006*). Future research should consider maturation-related variables such as bone structure, muscle mass, and tendon mechanical properties to better understand the implications of maturity and jump training on the causal interaction between joint kinematics and VGRF (*Werkhausen et al., 2018*). (v) Due to the relatively small size of our sample, it was not possible to differentiate between jumping sports and their interaction with sex. Therefore, these results should be taken with caution and cannot be extrapolated to other sports. Future studies will be required to confirm our results using different landing heights and comparing other sports. Finally, the causal empirical decomposition approach has some limitations that should be considered. Adding noise during the EEMD process, while necessary to improve the separability and orthogonality of the IMFs and to control spurious correlations (*Yang, Peng & Huang, 2018*, *2022*), may introduce challenges in interpreting the results (*Chang, Munch & Hsieh, 2022*). However, in our data, the direction of causality was robustly determined in 92% of cases. Future research should explore the development of enhanced methods to improve the understanding of causal interactions in biomechanics.

## CONCLUSIONS

Based on the causal decomposition approach, our results indicate that practicing jumping sports increases the causal effect of ankle kinematics on ground reaction forces in children. While non-jumper children rely more on the hip to modulate reaction forces, jumper children differ from non-jumpers by their greater use of the ankle joint. These findings could be used to develop specific training programs to improve landing techniques according to practice level, potentially helping to reduce the risk of injury in both athletes and non-athletes.

### Funding

Isaac Estevan was supported by the Conselleria de Educación, Investigación, Cultura y Deporte. Generalitat Valenciana, Spain (Grant number BEST 2019/074) for the study design, data collection and analysis, and preparation of the manuscript. This research was funded by the grant Healthy aging in industrial environment (program 4 HAIE CZ.02.1.01/0.0/0.0/16_019/0000798). This study has been supported by a research grant of the University of Valencia, Spain (UV-INV_EPDI-2015405). The funders had no role in study design, data collection and analysis, decision to publish, or preparation of the manuscript.

## Grant Disclosures

The following grant information was disclosed by the authors:

Conselleria de Educación, Investigación, Cultura y Deporte. Generalitat Valenciana, Spain: BEST 2019/074.

Healthy Aging in Industrial Environment: 4 HAIE CZ.02.1.01/0.0/0.0/16_019/0000798.

University of Valencia, Spain: UV-INV_EPDI-2015405.

## Competing Interests

The authors declare that they have no known competing financial interests or personal relationships that could have appeared to influence the work reported in this article.

## Author Contributions

- Carlos Cruz-Montecinos analyzed the data, prepared figures and/or tables, authored or reviewed drafts of the article, analysis tools, and approved the final draft.
- Isaac Estevan conceived and designed the experiments, performed the experiments, authored or reviewed drafts of the article, and approved the final draft.
- Jiri Skypala conceived and designed the experiments, performed the experiments, authored or reviewed drafts of the article, and approved the final draft.
- Claudio Tapia-Malebrán analyzed the data, authored or reviewed drafts of the article, analysis tools, and approved the final draft.
- Xavier García-Massó conceived and designed the experiments, performed the experiments, analyzed the data, prepared figures and/or tables, authored or reviewed drafts of the article, and approved the final draft.

## Human Ethics

The following information was supplied relating to ethical approvals (*i.e.*, approving body and any reference numbers):

University Valencia Ethics Committee.

## Data Availability

The raw data is available in the Supplemental Files.

## Supplemental Information

Supplemental information for this article can be found online at http://dx.doi.org/10.7717/peerj.18502#supplemental-information.

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
