# Peer review of "Effect of kinematics on ground reaction force during single-leg jump landing in children: a causal decomposition approach in jumpers and non-jumpers"

_PeerJ, doi:10.7717/peerj.18502_

## Round 0.1 · original submission · Major Revisions

Dear Authors:

The manuscript titled: "Jumping sports practice increases the causal effect of ankle kinematics on ground reaction force during single-leg jump landing in children" must be improved. Please attend to the reviews.

Regards

Dr. Manuel Jiménez

·

Basic reporting

Clear writing with strong supporting references. Structure of the article is appropriate with a hypothesis.

Experimental design

The experimental design is appropriate as long as the authors change the wording from "causal" to "causal empirical decomposition" throughout the articles, abstract, and title. The word "causal" by itself lends readers to believe the experimental design is temporal in nature (longitudinal).

Validity of the findings

The impact and novelty was not well described. The authors failed to show why this causal decomposition method introduces anything new beyond a linear relationship between kinematics and kinetics. It is an interesting and unique statistical approach, but without the "why this is important" developed.

Additional comments

Introduction
General: You do a good job reviewing the literature in the second paragraph, but I don’t see the point in understanding the causal relationship between kinetics and kinematics. Line 55-56 says understanding the causal interaction will help prevent injuries, but how? You need a paragraph describing how understanding the causal interaction between kinetics and kinematics can prevent injuries. Right now, to me, based on what you wrote, it does not matter for preventing injuries.

Lines 43-46: These studies did not “try”, I believe some of them did. Things like greater knee flexion relates to lower vGRF, for example. Perhaps a non-linear relationship is better/more accurate, but saying they tried sounds like they failes. But that’s not the case.

Lines 51-52: This sentence is actually very important because you talk about risk factors for injury. This sentence and type of discussion should be included in the new paragraph I proposed in the “general” section above. But it does not make sense in its current paragraph which discusses the relationship between various kinetics and kinematics.

Lines 59-60: You say “causal empirical decomposition” here. I suggest using this phrasing throughout the articles and title instead of simply saying “causal”. Saying “causal” by itself makes it sound like you had a study design that took into account temporal relationship between variables: X leads to Y, therefore X causes Y (an oversimplification, but that’s the general idea). However, you used a statistical approach, not wrong, but you need to be clear throughout the article that it is what you used.

Methods
General: Well done with the methods. Short, concise, but very clear.

Lines 92-94: How far away was the box from the force plates? Did the participants stick the landing, or land and jump again (e.g., step off box, land, jump into air, land)?

Lines 136-137: Please include effect size definition/ranges here. Then included effect sizes for all statistical outcomes.

Results
General: Reminder about effect sizes.

Lines 141-142: How did decide the cut-off for the 92%? In other words, earlier you described that an RCS value closer to 1 indicates kinematics cause vGRF, and 0.5 RCS meant to no causual relationship. Does that mean any value above 0.6 was considered kinematics causing vGRF and therefore included in the 92% results? What was the RCS cut-pff value for being included in the 92% results? How did you come up with this cut-off (used in prior literature)? If you used 0.6, is 0.6 really that different from 0.5?

Lines 142-143: Reminder about effect sizes.
Discussion
General: Similar to your introduction, you need a paragraph or two discussing what this means. Does this mean telling athletes to move through greater ankle range of motion will lead to lower vGRF? Does this mean we should prioritize ankle strengthening for jumping athletes, and hip strengthening for non-jumping athletes (control group)? Does this mean training programs should focus more on the ankle in general rather than vGRF for reducing injuries?

Lines 181-182: This sentence right here is important, because you mention that other studies have shown the exact same thing, a linear relationship between ankle ROM and vGRF. How does your study offer something new? What does the causal relationship mean?

·

Basic reporting

- The title is accurate and in line with the article's proposal.

Introduction: - I think it is important to mention that the more rigid jumping landing technique is a risk factor for some injuries, especially in children. The reference used is from a review with adults and it is not possible to identify whether the authors are referring to acute injuries or injuries caused by stress. What kind of injuries?

Introduction: - Falling techniques can have different impacts when it comes to bone, tendon and joint development. I think it is important to make this distinction, so as not to give the impression that “jumps are bad”, but rather that rigid landing strategies can be harmful in the long term.

Discussion: - In the discussion, the authors do not comment on the specifics of the participants’ sports. Volleyball and gymnastics have completely different jumping demands in terms of takeoff and landing, implying different self-awareness due to training.

Experimental design

- Regarding the inclusion criteria, how many times per week and what level of training did the children perform? Was there physical preparation or extra training?

- At 10 years old on average, it is possible to have girls approaching their growth peak, which means a greater hip bone structure, which could eventually be affecting the results. Would the authors be willing to apply a maturity offset equation to regulate the results?

- Although the approach provided tries to mitigate some of the criticisms raised about the non-linear method based on the causal empirical decomposition proposed by Yang et al., such as the separability of IMFs and the control of added noise, the criticisms about the robustness of the instantaneous phase analysis and the statistical validity of causal inferences may still apply. Therefore, it is essential that the authors acknowledge these limitations or answer about the strategy to mitigate these critical.

Chang, C. W., Munch, S. B. & Hsieh, C. H. Comments on identifying causal relationships in nonlinear dynamical systems via empirical mode decomposition. Nat. Commun. https://doi.org/10.1038/s41467-022-30359-8 (2022).

- The normal distribution was tested but the results were not reported. Since the sample is small, I suggest reporting them.

Validity of the findings

- The Shapiro-Wilk test is a good choice for testing the normality of data, but if the data are not normally distributed, non-parametric methods should be considered. The study does not mention what was done if normality was not met.

- Can the result for the control group be considered as important as that for the intervention group? If so, I suggest another approach in the conclusion and even in the title.

- The choice of the chi-square test to compare the distribution of the direction of causality is appropriate. However, there are no details about the sample size being large enough to guarantee the validity of the chi-square test (minimum of 5 cases in each box of the contingency table).

- Mixed ANOVA is appropriate for comparing the effects between factors (joints and groups). However, the approach depends on the assumption of sphericity, which should be verified (e.g., Mauchly's test). If sphericity is not met, corrections such as Greenhouse-Geisser should be applied. How did the authors proceed with this analysis?

Reviewer 3 ·

Basic reporting

Overall, the data are not comprehensive enough to support the conclusions of the article. The experimental design and statistical methods need to be improved. The research group is also vague, and it is difficult to tell which group of children it is.

Experimental design

The experimental design is not rigorous enough.

Validity of the findings

no comment

Additional comments

The manuscript revealed that jumping sports increase the causal effect of ankle kinematics on ground reaction forces. There are some issues to be discussed more before being published. The comments are as follows.

Introduction
Line 72-73 The description “This study may help to understand how jump practice changes the interaction between kinematics and kinetics” may not be appropriate. Please describe the research purpose in a more accurate way.

Materials and Methods
Line 78-79 The athletic ability of people of different genders or ages varies greatly. It would be better to provide detailed information on the age and gender of different groups to make research objects clear.

Results
Line 140 Quantitative data for average sagittal joint angles, VGRF and IMF are not shown in Figure 1. It is recommended to have quantitative graphs rather than raw data represented.

Figure 3 is just a schematic diagram and does not show any data. Overall, the data is not comprehensive enough to support the author's conclusion.

Shaping of the manuscript
English must be revised. Some sentences need modifications to be comprehensible

---

## Round 0.2 · Minor Revisions

Dear Authors:

I appreciate your patience. Could you please attend to this minor revision

Regards

Dr. Manuel Jiménez

·

Basic reporting

Clear writing and references are properly utilized.

Experimental design

All good.

Validity of the findings

The impact is well address and the novelty is unique.

Additional comments

General: Nice job on this updated manuscript. Minor comments are below. Good work!

Abstract
General: Please provide the actual effect size rather than just saying “large effect size”.

Introduction
General: Well written introduction. Good job! Thank you for taking my notes and comments into consideration.

Methods
General: Thank you for addressing my concern about the RCS cut off. I see you mentioned that the “largest” RCS value was used for analysis. And you provided the minimum in your results (Line 497). I appreciate the clarity.

Results
General: Well done.

Discussion
General: When you write “large effect size” please also show the actual effect size. Keep the words if you want, but provide the values as well.

·

Basic reporting

Dear author, thank you for carefully reviewing the article and the reviewers' reports. In my opinion, the manuscript is ready for publication, because all my comments have been addressed (with changes or assuming limitations).

Experimental design

The changes to the methods section clarified many points that were unclear or unclear. The careful writing of the statistical analysis makes it easier for the reader to understand all the steps of the analysis in detail.

Validity of the findings

no comments

Additional comments

I recommend the publication.

Reviewer 3 ·

Basic reporting

The revised article follows a logical flow, with clear transitions between sections.

Experimental design

The hypothesis of this research is clearly stated.

Validity of the findings

The methodology is clear, logical, and adequately described.

Additional comments

No comments.

---

## Round 0.3 · accepted · Accept

Dear Co-Authors:

Thank you for providing this new version. Your manuscript titled: Effect of kinematics on ground reaction force during single-leg jump landing in children: a causal decomposition approach in jumpers and non-jumpers - has been Accepted for publication.

Congratulations, and thank you for choosing PeerJ Journals.

Dr. Manuel Jiménez